# Single domain spectroscopic signatures of a magnetic kagome metal

L. Plucinski [1,2] ✉, G. Bihlmayer [3], Y. Mokrousov [3,4], Yishui Zhou [5], Yixi Su [5], J. D. Denlinger [6], A. Bostwick [6], C. Jozwiak [6], E. Rotenberg [6], D. Usachov [7] & C. M. Schneider [1,8,9]

Magnetic kagome metals host complex electronic states and real-space magnetic textures, but their small and temperature-dependent magnetic domains make experimental access difficult. Here we show that micro-focused circular-dichroic photoemission spectroscopy enables spectroscopic access to individual magnetic domains in the kagome metal $DyMn_6Sn_6$ at low temperature. By tuning to element-specific electronic states, we image domain contrast associated with Dy $4f$ levels and detect corresponding signatures from Mn core states. The energy dependence of the dichroic response is consistent with modeling and indicates ferrimagnetic alignment between Dy and Mn local moments. Measurements of Mn $3d$-derived valence bands, supported by first-principles calculations, reveal features related to orbital magnetization. These results establish element- and orbital-resolved spectroscopy of single magnetic domains and enable studies of magnetic textures and electronic structure in complex magnetic quantum materials.

Electronic structure of the kagome lattice features Dirac cones, flat bands, and Van Hove singularities. In materials research, a kagome lattice is often realized as built into the three-dimensional lattice, with numerous such compounds studied featuring topological and correlated phases[1]. Among such various materials, the $RT_6Sn_6$ compounds, where R is the $4f$ element, and T the transition metal, have attracted considerable attention due to exotic signatures in tunneling spectroscopy[2,3], quantum oscillations[4], as well as Dirac cones and flat bands[5]. Interactions between $4f$ and $3d$ ions have often been studied in relation to permanent magnets (e.g., $Nd_2Fe_{14}B$) applications, and the $RMn_6Sn_6$ family has been studied in some detail[6–8] due to its diverse magnetic phases, where different R elements lead to differently oriented easy magnetization vectors[9]. Another important property of a kagome lattice is the predicted existence of loop-currents[10,11] and strong orbital magnetism[12], potentially connected to superconductivity and the orbital Hall effect in some kagome metals. Within

the tight-binding picture, orbital magnetization stems either from on-site magnetic moments, characterized by $m_l$ quantum numbers, or from the intersite hopping loops[10], with CD-ARPES being sensitive to both processes.

Here, we show that properties of a single magnetic domain can be probed in the magnetic kagome metal $DyMn_6Sn_6$ using micro-focused circular-dichroic angle-resolved photoemission (μ-CD-ARPES). We resolve magnetic domains in samples cryo-cooled to 20 K through robust signatures in the Dy $4f$ multiplet region, and detect smaller but clear domain-related signatures in the Mn $3p$ core-level region. Comparing Dy $4f$ and Mn $3p$ spectra with our modeling based on the Hartree–Fock method, we identify ferrimagnetic alignment of the Dy and Mn local moments. We further demonstrate the response of the Mn $3d$-dominated valence bands to CD-ARPES and relate it to first-principles calculations, revealing signatures consistent with non-vanishing orbital magnetization in a kagome metal.

[1]Peter Grünberg Institut (PGI-6), Forschungszentrum Jülich GmbH, Jülich, Germany. [2]Institute for Experimental Physics II B, RWTH Aachen University, Aachen, Germany. [3]Peter Grünberg Institut (PGI-1), Forschungszentrum Jülich and JARA, Jülich, Germany. [4]Institute of Physics, Johannes-Gutenberg University Mainz, Mainz, Germany. [5]Jülich Centre for Neutron Science (JCNS) at Heinz Maier-Leibnitz Zentrum (MLZ), Forschungszentrum Jülich, Garching, Germany. [6]Advanced Light Source, Lawrence Berkeley National Laboratory, Berkeley, CA, USA. [7]Donostia International Physics Center (DIPC), Donostia-San Sebastian, Spain. [8]Fakultät für Physik, Universität Duisburg-Essen, Duisburg, Germany. [9]Physics Department, University of California, Davis, CA, USA. ✉e-mail: l.plucinski@fz-juelich.de

## Results

Figure 1 shows magnetic domains in the DyMn₆Sn₆ single crystal cleaved under ultra-high vacuum, probed by μ-CD-ARPES. The X-ray photoemission (XPS) spectrum and the experimental geometry are shown in Fig. 1a, b, respectively. The Dy $4f$ multiplet exhibits a complex structure[13], with an isolated feature stemming predominantly from $^7F$ terms, indicated in Fig. 1a. Figure 1c–e show micrographs of the sample surface averaging over the Dy $4f$ $^7F$ feature, performed by scanning the photon beam of ≈2 μm diameter[14] over the sample surface. Figure 1c shows the sum of the intensities $I_+$ and $I_-$, measured with circularly polarized $C_+$ and $C_-$ light, respectively, at $hv$ = 300 eV, with a high intensity region enclosed by the yellow dashed line. Normalized difference $(I_+ - I_-)/(I_+ + I_-)$, here termed the CD magnitude, is shown in Fig. 1d, e for $hv$ = 300 and 140 eV, respectively, where in both cases the same CD pattern is revealed within the high intensity region of Fig. 1a, with CD reaching ≈90% in Fig. 1d and ≈50% in Fig. 1e. Similar experiments performed for Mn $3p$ are shown in Fig. 1f–h where the micrographs in Fig. 1f, g are related to the low and high binding energy features of the Mn $3p$, respectively, as indicated in Fig. 1h.

Detailed analysis of the Dy $4f$ spectral region at two representative areas $A$ and $B$ indicated in Fig. 1d is shown in Fig. 2, with the emission angles along the blue fan shown in Fig. 1b. Figure 2a shows the sum of the measurements performed with $C_\pm$ light at $A$ and $B$, Fig. 2b shows the difference of the intensities with $C_+$ and $C_-$, $I_{+A} - I_{-A}$, at area $A$, and Fig. 2c shows the difference $I_{+B} - I_{-B}$ at area $B$. All three maps, Fig. 2a–c, show strong emission-angle-dependent modulations, which we theoretically model by assuming the kagome-terminated surface depicted

in Fig. 2d and a magnetization vector $\mathbf{M}$ at 45° with respect to the surface normal, projected along the $\Gamma K$ azimuthal direction indicated in Fig. 2e (see Supplementary Material for a detailed analysis of alternative orientations of $\mathbf{M}$).

Figure 2f, g show our theoretical maps at $\mathbf{M}$ and $-\mathbf{M}$, where electronic correlations are simulated within the Hartree–Fock formalism, and the propagation is performed using EDAC[15] (see Section "Photoemission calculations" for details). The correspondence between the experimental and theoretical maps regarding the signs of the intensity differences is quantitative, which provides a strong evidence that opposite signs of CD in the Fig. 1d, e are related to magnetic domains at the quasi-kagome-terminated surface (Fig. 2d).

Figure 2h, i show experimental and theoretical energy distribution curves of $I_{+A} + I_{-B}$ (red curves) and $I_{+B} + I_{-A}$ (blue curves), integrated over the full experimental angular fan (see Fig. 1b). These curves aim to show the magnetic contribution to the signal, while canceling out dichroic signals due to the final state scattering[16]. The agreement between experiment and theory is again quantitative.

In Fig. 2j we demonstrate that plotting a linear combination of signals $(I_{A+} - I_{A-}) - (I_{B+} - I_{B-})$, that is the difference between red and blue curves in Fig. 2h, one can remove most of the nonmagnetic modulations and obtain a map where the sign of the dichroic signal does not depend on the emission angle. Finally, the remarkable agreement between the experiment and theory is again confirmed in Fig. 2k, where the angle-integrated curve of Fig. 2j is compared to the theoretical one.

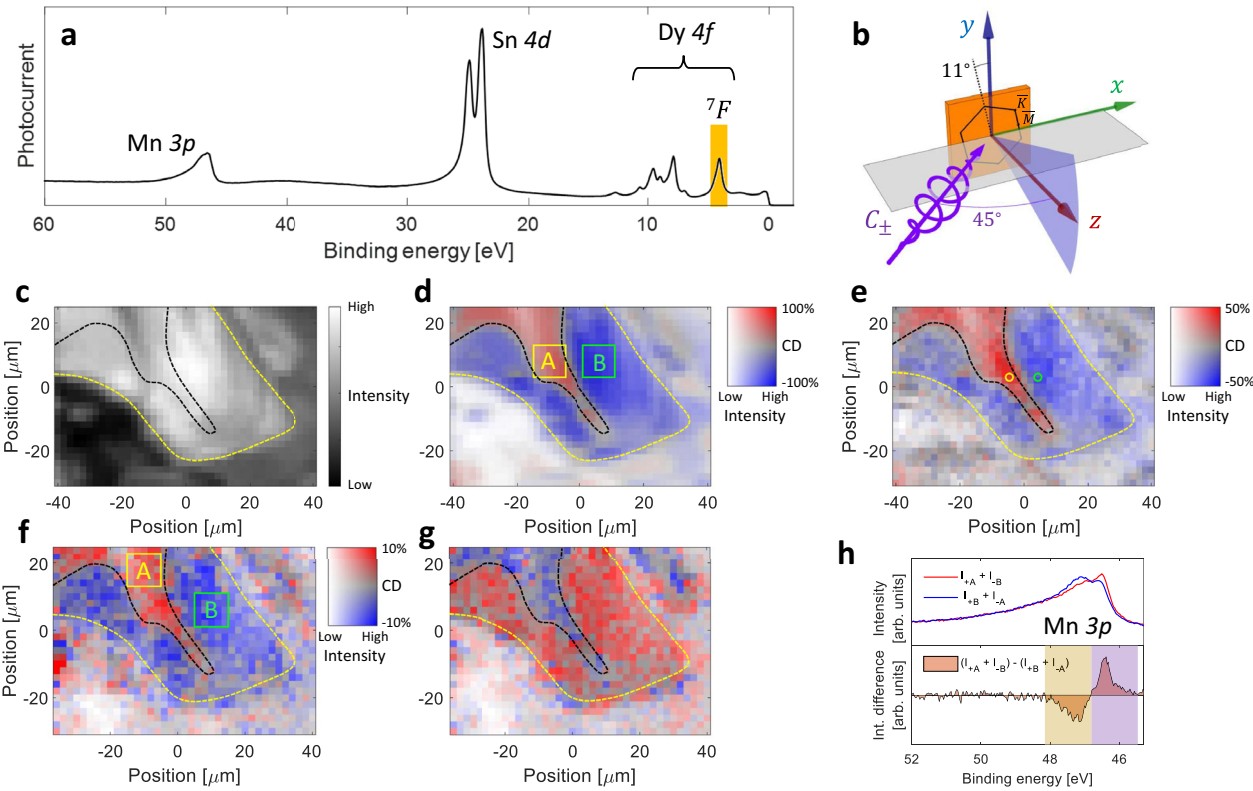

**Fig. 1 | Single-domain magnetic contrast in DyMn₆Sn₆ revealed by μ-CD-ARPES.** **a** XPS spectrum from cleaved DyMn₆Sn₆ surface measured at $hv$ = 300 eV. **b** Experimental geometry. **c** Micrograph of the sample surface taken by scanning the sample by the $hv$ = 300 eV photon beam of the ≈2 μm diameter while measuring the signal of Dy $4f$ $^7F$ feature, as highlighted in (**a**), sum of the intensities measured with $C_\pm$ light, $I_+ + I_-$. **d** Circular-dichroic (CD) signal $(I_+ - I_-)/(I_+ + I_-)$ plotted according to the colormap that shows both the CD strength and the photoemission intensity. Yellow and green rectangles indicate regions of domains $A$ and $B$ used in further

analysis. **e** Same as **d** but at $hv$ = 140 eV, and with the 2D colormap saturated at CD of 50% as indicated. **f, g** Micrographs measured over the low (violet) and high (beige) binding energy regions as indicated in the lower panel of (**h**). Red and blue curves in (**h**) show the $I_{+A} + I_{-B}$ and $I_{+B} + I_{-A}$ as well as their difference, where $A$ and $B$ refer to boxes in (**f**). Colormap in **f, g** has been saturated to CD of 10% as indicated. CD strengths have been calculated with linear backgrounds of Dy $4f$ $^7F$ and Mn $3p$ subtracted.

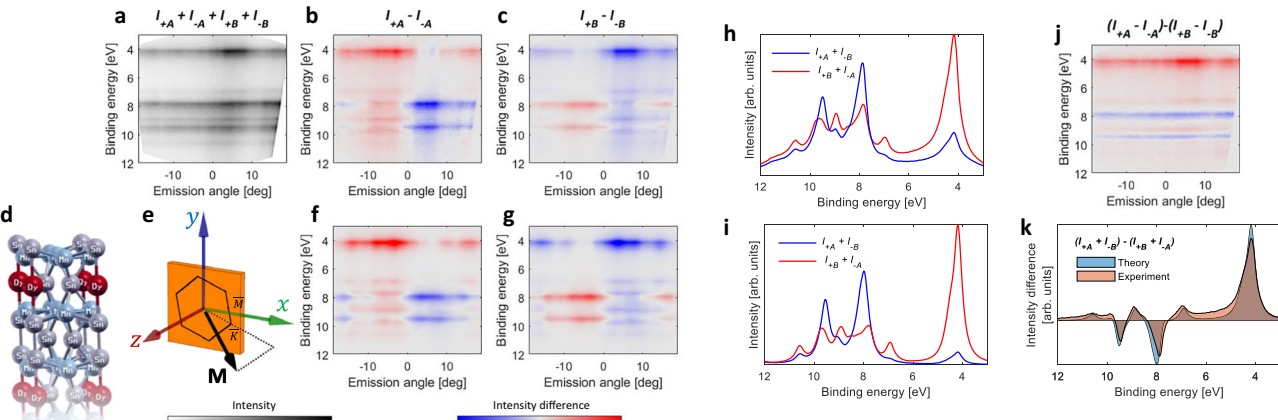

**Fig. 2 | Experimental and theoretical energy-momentum maps for the Dy 4$f$ multiplet region at $h\nu = 300$ eV, taken at regions $A$ and $B$, as indicated in Fig. 1d.** **a** $I_{A-} + I_{B-} + I_{A+} + I_{B+}$. **b** $I_{A+} - I_{A-}$ at domain $A$. **c** $I_{B+} - I_{B-}$ at domain $B$. **d** Schematic illustration of the kagome termination used in our theoretical modeling. **e** Orientation of the magnetization vector **M** used in the theoretical calculation,

compare to Fig. 1b. **f, g** Theoretical modeling for **M** and -**M** aimed to simulate (**b, c**). **h** Angle-integrated spectra of $I_{A+} - I_{B-}$ and $I_{B+} - I_{A-}$. **i** Theoretical simulation of (**h**). **j** Difference between maps (**b, c**). **k** Angle integrated spectrum of **j** (beige filling) together with the theoretical simulation (blue filling).

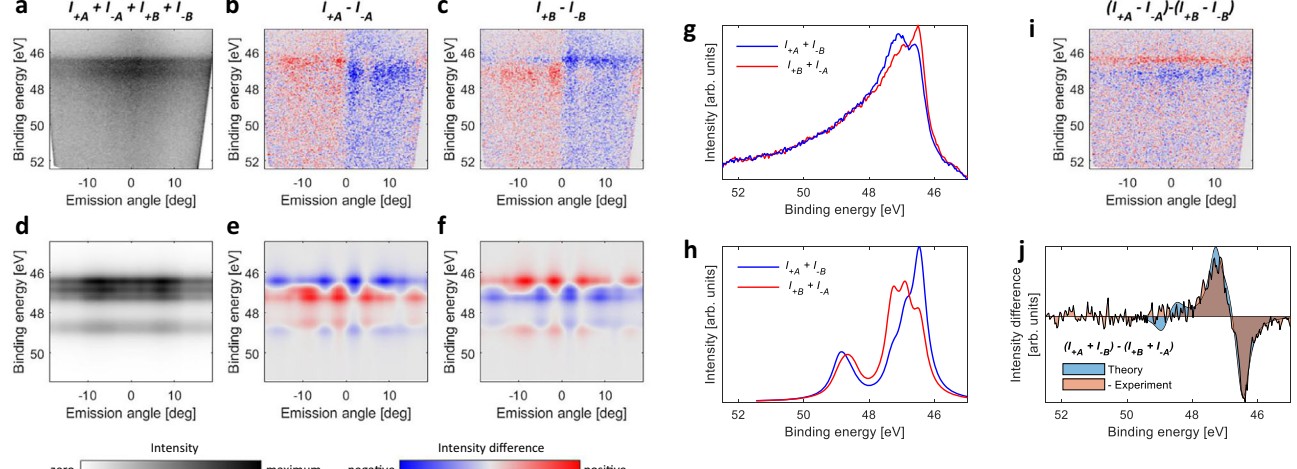

**Fig. 3 | Experimental and theoretical energy-momentum maps for the Mn 3$p$ multiplet region at $h\nu = 300$ eV, taken at regions $A$ and $B$, as indicated in Fig. 1f.** **a** $I_{A-} + I_{B-} + I_{A+} + I_{B+}$. **b** $I_{A+} - I_{A-}$ at domain $A$. **c** $I_{B+} - I_{B-}$ at domain $B$. **d** Theoretical simulation of (**a**). **e, f** Theoretical modeling for **M** and -**M** aimed to simulate (**b, c**).

**g** Angle-integrated spectra of (**b, c**). **h** Theoretical simulation of (**g**). **i** Difference between maps (**b, c**). **j** Sign-inverted angle integrated spectrum of **i** (beige filling) together with the theoretical simulation (blue filling).

Notably, one can observe a tendency of the dichroic signal in Fig. 2b, c, where the CD sign is primarily positive (red color) for negative emission angles and negative (blue) for the positive emission angles. This is an expected behavior in CD-ARPES maps[17], stemming from atomic photoionization CD profiles and the Daimon effect[16].

Figure 3a–c shows the experimental and Fig. 3d–f theoretical maps for the Mn 3$p$ spectral range, related to the regions $A$ and $B$ in Fig. 1f. Comparison of the intensity maps in Fig. 3a, d reveals predominant double-peak experimental structure, and a more complex theoretical map with at least 4 visually present peaks. Experimental intensity difference maps for $C_{\pm}$ light in Fig. 3b, c reveal angle-dependent structures as well as dichroic signal of the background, changing the sign for negative and positive angles. This character of the background is the same for domains $A$ and $B$, therefore, it is not related to the magnetism, and we associate it primarily with the inelastic electron background related to dichroism from the valence-band region. This background is not present in our theoretical maps,

Fig. 3e, f, because they do not take into account inelastic scattering. Other than the lack of the background, theoretical maps in Fig. 3e, f show structures similar to the experimental ones, however, exhibiting sign reversal, demonstrating ferrimagnetic order of the Dy and Mn local moments. In order to cancel the influence of the background, in Fig. 3g we plot $I_{A+} + I_{B-}$ (red curve) and $I_{B+} + I_{A-}$ (blue curve), and compare it to the theoretical simulation in Fig. 3h. One can again see the dichroic signal sign reversal, as compared to the Dy 4$f$. With this reversal taken into account, there is an agreement between experiment and theory, however, numerous features predicted by our theory appear smeared out in the experimental curve. In Fig. 3i, we plot the linear combination $(I_{A+} - I_{A-}) - (I_{B+} - I_{B-})$, demonstrating that this way the contributions due to final state scattering can be suppressed, resulting in the angle-independent sign of the dichroic signal. Integrating Fig. 3i over angles and reversing the sign, one obtains the curve in Fig. 3j (beige fill), which, for the main features, is in quantitative agreement with the theoretical one (blue fill).

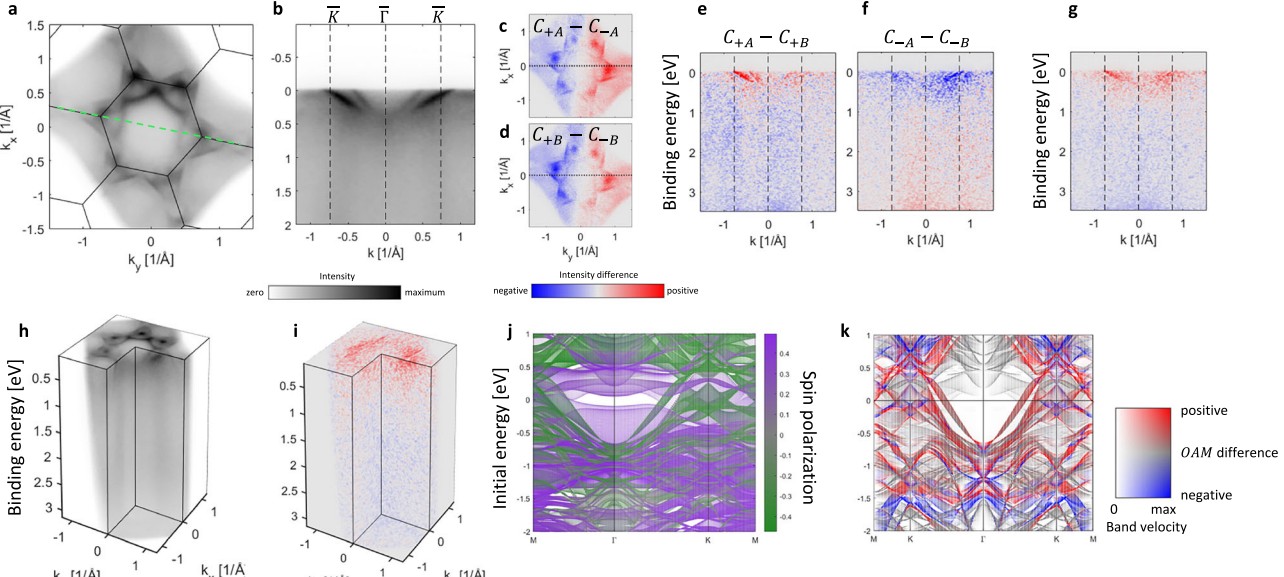

**Fig. 4 | Magnetic-domain-resolved valence-band response and calculated orbital-angular-momentum signatures. a** Fermi surface measured using $p$-polarized light at $hv = 140$ eV. **b** Energy-momentum map along the green dashed line in (**a**), representing the $\overline{MK\Gamma KM}$ trajectory. **c, d** CD-ARPES at the Fermi surface at spots $A$ and $B$ of Fig. 1e. **e** Energy-momentum map for the same trajectory as (**b**), showing the difference between measurements at spots $A$ and $B$ measured with the $C_+$ light. **f** Same as (**e**) but for $C_-$ light. **g** Difference between (**e**, **f**). **h** 3D representation of the data set related to (**a**, **d**). **i** 3D representation of the data related to (**g**). **j** Projected bulk band structure with the color indicating spin character at Mn sites. **k** Projected bulk band structure for the trajectory of **b**, **e**, **f**, **g** with the color indicating OAM difference between $\pm\mathbf{M}$ (Fig. 2e) of Mn sites along the quantization axis defined by the light incidence, see Fig. 1b.

Figure 4a, b shows the familiar Fermi surface and energy-momentum map of the RMn$_6$Sn$_6$ compound family. Figure 4c, d show the related, visually alike, CD-ARPES at the two domains, which suggests that these CD signals originate primarily from the Daimon effect (see also Section IV of the Supplementary Material). In order to filter out magnetic contributions to the CD signal, we analyze energy-momentum maps where we plot the difference of the spectra taken at two domains, $I_{+A} - I_{+B}$ in Fig. 4e and $I_{-A} - I_{-B}$ in Fig. 4f. The magnetic origin of the signal is confirmed by the sign reversal in the overall character of these maps, and is further confirmed in Fig. 4g, where the difference of Fig. 4e, f is shown. Figure 4h, i show 3D impressions of our ARPES data related to Fig. 4a, b, g, respectively, showing that the map in Fig. 4g is the representative trend over the entire probed ($E_B$, $k_x$, $k_y$) space.

In order to understand these results, we performed ab initio electronic structure calculations with the representative results shown in Fig. 4j, k. We focus on Mn band characters since they dominate near the Fermi level, and furthermore, the cross-section of Mn 3$d$ is much higher than any other contribution (see Supplementary Material SIII for details), therefore, our experiments in Fig. 4 probe predominantly the kagome-arranged layer of Mn atoms. With the details analyzed in the Supplementary Material, our results suggest that the two dispersive branches in Fig. 4b originate from the projected bulk band structure and are primarily related to the Mn minority spin channel (green color in Fig. 4j). Importantly, circular light provides sensitivity to the OAM and not directly to the spin character. Therefore, in order to understand the results of Fig. 4e–g, we analyze the difference between the OAM characters for the opposite magnetizations of the sample. Photoemission matrix element for $C_\pm$ light can be written as $\langle \psi_f | (\varepsilon_{x'} \pm i\varepsilon_{y'}) \cdot \mathbf{p} | \psi_i \rangle$, where in the coordinate system ($x'$, $y'$, $z'$) the $z'$ axis is along the incident light and should be used as a quantization axis for the determination of the OAM. Following our experimental geometry, in Fig. 4k we visualize the difference between such calculated OAM for $\pm\mathbf{M}$, $OAM_{+\mathbf{M}} - OAM_{-\mathbf{M}}$, for bands along the trajectory of Fig. 4b. There,

using the 2D colormap, we highlight steeply dispersing bands, since they appear most prominent in the experimental map in Fig. 4b. One can see that, in agreement with the experiment in Fig. 4g, dispersing bands near the Fermi level exhibit consistent sign, suggesting that our experiment indeed enables spectroscopic access to the OAM properties of a magnetic kagome metal. This interpretation is strict only in the case of the central potential approximation, and we discuss the added complexities for periodic solid surfaces in the Supplementary Material. In our work, the connection to the magnetic properties is possible only through access to a single domain, with the additional benefit of having two domains oriented antiparallel. This allows to filter out the contributions to the signal that do not stem from magnetism, and can eliminate contributions related to the Daimon effect. Importantly, as typical in RMn$_6$Sn$_6$ materials, well-defined quasiparticle bands are present only up to $\approx 0.5$ eV below the Fermi level (Fig. 4b), which limits the range that can be compared to our calculations.

Moreover, there exist an important difference between our approach and the X-ray circular dichroism (XMCD)[18]. XMCD predominantly probes the spin contribution to the magnetization through filtering of the core-level emission by the strongly spin-polarized states immediately above the Fermi level (although orbital magnetization can also be probed[19,20]), whereas CD-ARPES is only sensitive to the OAM. Since by probing the two anti-parallel domains the magnetization is being reversed, the results in Fig. 4g, i are related to magnetism, and therefore, through the OAM sensitivity of CD-ARPES, they represent an insight into the orbital magnetization of DyMn$_6$Sn$_6$, and in particular its contribution from the kagome-arranged Mn atoms. By accessing the OAM in a magnetic system, our study opens new avenues for characterizing the wave function properties that define the quantum geometric tensor in solids[21,22]. Going beyond the atomic picture leads to a possible further distinction between atomic-like and Berry phase-related contributions to the orbital magnetization[20,23], a topic that deserves future attention.

## Methods

### Sample preparation

Single crystals of $DyMn_6Sn_6$ were grown using the Sn self-flux method[2]. A Dy lump, Mn pieces, and Sn shots were loaded into a crucible in a glove box with a molar ratio of Dy: Mn: Sn = 1: 6: 22. The crucible was sealed in a glass tube with quartz wool under high vacuum. The sealed tube was heated to 1050 °C over 10 h and held at this temperature for 10 h. It was then slowly cooled to 650 °C at a rate of 2 °C/h, followed by centrifuging to separate the crystals from the Sn flux. Then, plate-shaped crystals with a shiny surface were obtained. The single crystal sample has been cleaved under ultra-high-vacuum using the ceramic post technique.

### Photoemission measurements

Photoemission measurements were performed at the NanoARPES branch of the MAESTRO beamline at the Advanced Light Source. A focusing capillary was used to obtain a beam spot with a diameter below 2 μm[14]. The overall energy resolution was approximately 0.2 eV at $h\nu = 300$ eV and 0.1 eV at $h\nu = 140$ eV, with an angular resolution of 0.1°. The sample was maintained at 20 K during the measurements, and the pressure in the analyzer chamber was below $5 \times 10^{-11}$ mbar. No sample drift or changes in the magnetic domain structure were observed over the ~12 h measurement period.

### Photoemission calculations

The matrix element describing electron emission from the atom $a$ being in the ground state $|gJM_J\rangle$ with the total momentum $J$ and its projection $M_J$ is given by

$$\langle \vec{k}m_s\beta|T|gJM_J\rangle = \sum_{jm_jJ'M_jlmq} \frac{\langle \beta J_f, klsj{:}J'||D||gJ\rangle}{\sqrt{2J'+1}}$$

$$\varepsilon_q C_{JM_J,1q}^{J'M_J'} C_{lm,sm_s}^{jm_j} C_{J_fM_{jf},jm_j}^{J'M_J'} i^{-l} e^{i\delta_l} \psi_{a,lm}(\vec{k}), \qquad (1)$$

where $\beta$ is the final state of the ionized atom with the total momentum $J_f$, $\vec{k}$ is the momentum of the photoelectron with the spin projection $m_s$, $T$ is the PE transition operator, $C$ denotes the Clebsch-Gordan coefficients, $\delta_l$ is the phase of the partial photoelectron wave with the orbital momentum $l$, $\psi_{a,lm}$ is the amplitude of the partial photoelectron wave emitted from the atom $a$ and scattered on the surrounding atomic environment (see ref. 24), and $\varepsilon_q$ are the spherical components of the photon polarization vector.

The reduced matrix element of the dipole operator $\langle \beta J_f, klsj{:}J'||D||gJ\rangle$ was calculated using the atomic multiplet theory. The Hamiltonian parameters for Dy were taken from ref. 25. The half-width of Dy $4f$ PE lines was set to 0.2 eV. For the Hamiltonian of Mn $3p^6 3d^5$ and $3p^5 3d^6$ configurations we used the SOC radial integrals taken from the Hartree–Fock calculation: $\zeta(3d) = 0.046$ eV and $\zeta(3p) = 0.79$ eV; to account for screening, we notably reduced the calculated Slater integrals and used the following empirical values in eV: $F^2(3d, 3d) = 7$, $F^4(3d, 3d) = 5$, $F^2(3p, 3d) = 8.6$, $G^1(3p, 3d) = 11.2$, $G^3(3p, 3d) = 3.5$. We neglected the crystal field effects and the energy splitting of states due to magnetic ordering. Polarized ground states of Dy and Mn atoms were modeled by rotating the state with $M_J = J$ to obtain the desired direction of the total moment. The half-width of the Mn $3p$ PE lines was calculated as the rate of the $3p^5 3d^5 \rightarrow 3p^6 3d^4$ decay process plus 0.25 eV (which stands for all other contributions to the lifetime).

The amplitudes $\psi_{a,lm}$ were calculated with the EDAC program[15]. We used differently terminated cylindrical clusters of $DyMn_6Sn_6$ with the radius and height of 30 Å. For the modeling of Dy spectra, we considered two emitters in the two near-surface Dy layers, while for Mn, we used twelve emitters from the four layers.

### Initial state calculations

The density functional theory calculations were performed within the generalized gradient approximation[26] using the full-potential linearized augmented plane-wave (FLAPW) method[27] as implemented in the FLEUR-code[28]. Spin-orbit coupling was included self-consistently[29] with the spin-quantization axis tilted 45 (135) degree from the z-axis (0001) in the x-direction ($\bar{1}100$). The muffin-tin radii were 1.33 (1.48) Å for the Sn and Mn (Dy) atoms. The self-consistent calculations were performed with a $9 \times 9 \times 6$ k-point grid and a LAPW cutoff of 7.37 Å$^{-1}$. A Hubbard $U$ of 6.6 eV was added to the $4f$ states of the Dy. The structure was taken from ref. 30.

## Data availability

The data that support the findings of this study are available from the corresponding author upon request.

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

## Acknowledgements

Y.M. acknowledges support by the Deutsche Forschungsgemeinschaft (DFG) in the framework of TRR 288/2-422213477 (Project B06), and by the EIC Pathfinder OPEN grant 101129641 "OBELIX". G.B. gratefully acknowledges the computing time granted through VSR on the supercomputer JURECA-DC at Forschungszentrum Jülich. This research used resources of the Advanced Light Source, which is a DOE Office of Science User Facility under contract no. DE-AC02-05CH11231. This work was supported by the Deutsche Forschungsgemeinschaft (DFG, German Research Foundation) under Germany's Excellence Strategy-Cluster of Excellence Matter and Light for Quantum Computing (ML4Q) EXC 2004/1-390534769.

## Author contributions

L.P., A.B., C.J., and E.R. performed the ARPES experiments. D.U. performed the photoelectron diffraction and related theoretical calculations. G.B. performed the electronic structure calculations. Y.M. contributed to the theoretical interpretation. J.D.D. contributed to the experimental interpretation. Y.Z., Y.S. synthesized the samples and contributed to the analysis of magnetic properties. C.M.S. supervised the project.

## Funding

## Competing interests

The authors declare no competing interests.

## Additional information

**Supplementary information** The online version contains Supplementary material available at https://doi.org/10.1038/s41467-026-71924-9.

