## [Transparent Peer Review File · Nature Communications]

Single domain spectroscopic signatures of a magnetic Kagome metal

Corresponding Author: Dr Lukasz Plucinski

Version 1:

Reviewer comments:

Reviewer #1

(Remarks to the Author)

The manuscript "Single domain spectroscopic signatures of a magnetic Kagome metal" by L. Plucinski et al, examines single domains of the magnetic Kagome metal DyMn6Sn6 via micro-ARPES and circularly polarized light. The circular dichroic responses of both Dy 4f and Mn 3p orbitals, as well as that of the Fermi-level states, are investigated, and the magnetic contribution is isolated by examining the response difference between the two magnetic domains. Theoretical calculations accurately capture the photoemission details and quantitatively agree with the data, allowing for a further isolation of the magnetic contribution from the total photoemission signal.

In our view, the main advances in this work are: (1) Technologically: the use of CD-ARPES within a μ -ARPES setup, demonstrating the ability to identify different magnetic domains in a Kagome-magnet, and (2) the analysis using the domain-specific and polarization-specific data to isolate magnetic contribution to the signal. Which finally led to (3) The ability to extract the magnetic-OAM related to orbital magnetization and better understanding of Kagome magnets.

The manuscript is well-written, and the analysis of both experimental data and theoretical results is sound and well-presented. Demonstrating the ability to extract magnetic information by considering two domains would certainly contribute to the fields of CD-ARPES and Kagome materials. There are however a few points that need to be addressed:

(1) In the introduction, it is stated in regards to the magnetization axis: "DyMn6Sn6, which exhibits an easy axis at $\approx 45^\circ$ with respect to the c -axis [9]". However reference [9] says that this material has an easy-cone magnetization anisotropy, and not an easy axis. In the manuscript the authors assume an easy-axis, and further assume the magnetization axis projection is along the Gamma-K axis. Is there a justification for this assumption? If not, how would the calculations change for a different magnetization direction? Furthermore, the assumption that the two domains have opposite magnetization is not clear in an easy-cone scenario, as there are more degrees of freedom than just sign inversion. Why is this assumption warranted?

(2) There are two magnetization domains A and B, however the authors chose different integration regions for different analyses / binding energies, as marked in fig.1 (d) and (f). It is unclear why different integration regions are picked, or further - why isn't the whole area of a given domain integrated over for better signal. This needs to be explained, as it raises concerns of hidden spatial variability in the data.

(3) The crystal orientation is rotated by 11 degrees compared to the analyzer's slit, as depicted in the experimental geometry shown in Fig. 1(b). Such orientation leads to substantial matrix element effects that conventional CD-ARPES experiments aim to mitigate by aligning the sample's high-symmetry direction with the slit. The two CD-Fermi surface maps presented in Fig. 4(c-d) exhibit a distinct two-fold structure around $k_y=0$, rather than the three- or six-fold texture that could be expected from the crystal structure's symmetry, indicating a major contribution of matrix element effect (in addition to final-state scattering/Damion effect that the authors mention). The reason for choosing this sample orientation is unclear, especially since the authors go to great lengths to isolate the magnetic contribution from all the others. The effects of working with the plane-of-incidence of the light not along the mirror symmetry axis of the sample should be discussed.

Final minor suggestion: The red-to-blue color scale is used extensively throughout the manuscript and supplementary material, for related but distinct quantities. Choosing different colorscales for different quantities could improve the paper's readability.

Reviewer #2

(Remarks to the Author)

Reviewer #3

(Remarks to the Author)

This paper reports the magnetic properties of a Kagome lattice metal, DyMn₆Sn₆. By performing angle-resolved photoemission measurements of the Dy 4f and Mn 3p levels using a circularly polarized micro-focused beams with a diameter of less than 2 micrometer, the authors succeeded to reveal that the local magnetic moments of Dy and Mn exhibit ferrimagnetic ordering. The results obtained are interesting, but not surprising. Since even more interesting results could have been obtained with information on electron spin, I am wondering why the authors did not try to do spin-resolved photoemission measurements. Adding comments on this point would make the advantages of the method used by the authors in this study clearer.

Furthermore, detailed explanation on why does the signs of (I+A + I-B) and (I+B+ I-A) are opposite in the experimental result and theoretical one should be added, together with the reason of the difference in the percentage of CD when using different photon energies.

Minor points

Typo: On the first page, in the right column, "singatures" should be "signatures".

Reviewer #4

(Remarks to the Author)

The present manuscript has reported the study of the magnetic Kagome metal DyMn₆Sn₆ using high-resolution micro-focused circular-dichroic angle-resolved photoemission to probe its magnetic and electronic properties using combined experimental and theoretical techniques. The work can be very useful from experimental techniques point of view. I would definitely suggest this work to NCOMMS article. I still put one questions below whose entire descriptions not available.

Could you please elaborate the connecting techniques in supporting documents or in the main manuscript stating how ab-initio calculations were used for calculating photoemission matrix elements and which tool did you use?

Version 2:

Reviewer comments:

Reviewer #1

(Remarks to the Author)

We have read the detailed response and the new manuscript.

The authors have adequately answered ours and the rest of the referees questions. We recommend publication.

Reviewer #2

(Remarks to the Author)

Reviewer #3

(Remarks to the Author)

The authors have answered properly to all my comments, and I therefore recommend this paper for publication in the present form.

Reviewer #4

(Remarks to the Author)

After reading the reply from the authors to the previous reviewers, I conclude that such an experiment and theoretical analysis would definitely lead to more study on these materials and the topic. Ofcourse there will always be some space for analysis for future work. But the analysis the authors have made is very useful. I definitely would suggest for a publication in Nature Communication.

With this rebuttal letter we are submitting a revised manuscript main text and supplemental information. In these revised versions, we addressed the comments of the reviewers, in particular regarding the magnetic anisotropy (now addressed in detail in Supplemental Sec. SV) and data analysis over a larger sample surface area (now addressed in detail in Supplemental Sec. SVI). We also update the false colormap of selected figures, now all the panels related to spin polarization use the violet-green colormap. In all cases the main text has been augmented accordingly.

We explain the details *inline* below, following the comments of the reviewers.

Reviewer #1 (Remarks to the Author):

The manuscript "Single domain spectroscopic signatures of a magnetic Kagome metal" by L. Plucinski et al, examines single domains of the magnetic Kagome metal DyMn6Sn6 via micro-ARPES and circularly polarized light. The circular dichroic responses of both Dy 4f and Mn 3p orbitals, as well as that of the Fermi-level states, are investigated, and the magnetic contribution is isolated by examining the response difference between the two magnetic domains. Theoretical calculations accurately capture the photoemission details and quantitatively agree with the data, allowing for a further isolation of the magnetic contribution from the total photoemission signal.

In our view, the main advances in this work are: (1) Technologically: the use of CD-ARPES within a mu-ARPES setup, demonstrating the ability to identify different magnetic domains in a Kagome-magnet, and (2) the analysis using the domain-specific and polarization-specific data to isolate magnetic contribution to the signal. Which finally led to (3) The ability to extract the magnetic-OAM related to orbital magnetization and better understanding of Kagome magnets.

We would like to thank Reviewer #1 for their positive in depth assessment of our work.

The manuscript is well-written, and the analysis of both experimental data and theoretical results is sound and well-presented. Demonstrating the ability to extract magnetic information by considering two domains would certainly contribute to the fields of CD-ARPES and Kagome materials. There are however a few points that need to be addressed:

(1) In the introduction, it is stated in regards to the magnetization axis: "DyMn6Sn6, which exhibits an easy axis at $\approx 45^\circ$ with respect to the c-axis [9]". However reference [9] says that this material has an easy-cone magnetization anisotropy, and not an easy axis. In the manuscript the authors assume an easy-axis, and further assume the magnetization axis projection is along the Gamma-K axis. Is there a justification for this assumption? If not, how would the calculations change for a different magnetization direction? Furthermore, the assumption that the two domains have opposite magnetization is not clear in an easy-cone scenario, as there are more degrees of freedom than just sign inversion. Why is this assumption warranted?

The question of whether “easy cone” represents a physical arrangement of **M** at low temperatures in R166 compounds is currently under debate. Co-authors Yixi Su and Yishui Zhou have recently reported (Phys. Rev. Research 6, 043291 (2024) <https://doi.org/10.1103/PhysRevResearch.6.043291>) that in another Kagome compound DyV₆Sn₆ the moment is canted from the c-axis by approximately 20° and projected in-plane

along $\overline{\Gamma K}$, however, the in-plane anisotropy is very small. The older paper of Malaman et al., JMMM 202, 519 (1999), [https://doi.org/10.1016/S0304-8853\(99\)00300-5](https://doi.org/10.1016/S0304-8853(99)00300-5) does not really discuss the issue of in-plane anisotropy, it is possible that in these measurements it was difficult to distinguish between “easy cone” and “easy axis”. Our measurements have been performed at 20K, where small in-plane anisotropies can likely fix the magnetization direction.

Figure R1: Theoretical prediction for the approximate “easy cone” arrangement of \mathbf{M} , sum of 12 azimuthal orientations of \mathbf{M} (projected along all ΓM and ΓK directions), and at polar angle of 45° . The panel shows same as Fig. 2f of the main text but for “easy cone”.

Figure R1 shows our photoelectron diffraction simulation for the “easy cone”, arrangement (we sum 12 different azimuthal projections, assuming 45° polar angle). It is evident that this simulation does not correspond to the experimental data in Fig. 2b and 2h, therefore, it is an argument for the “easy axis” and against “easy cone”, in our sample at 20K.

Figure R2: Ab-initio calculations (FLEUR) for the bulk magnetic anisotropy in DyMn_6Sn_6 . Abscissa depicts the off-normal angle θ of the magnetization \mathbf{M} (the angle away from the c -axis), ordinate shows total energy differences for the different orientations of \mathbf{M} . The difference between blue and red curves is the azimuthal orientation. The blue curves show the case when magnetization is projected onto ΓK and the red curve onto ΓM direction. Right panels show the relation between the real and reciprocal in-plane vectors for this calculation.

Our choice of easy axis projection along $\overline{\Gamma K}$ and away from the light incidence (Fig. 2e) is based on simulating several possible \mathbf{M} directions, and choosing the one that exhibits the best match

to the experiment in Fig. 2**b,c,h**. This is now discussed in detail in Sec. SV and Figs. S5 and S6 of the revised Supplement. In particular, Fig. S6 shows simulations for **M** projected along all six $\overline{\Gamma K}$ directions.

Furthermore, we performed DFT total energy calculations for different azimuthal orientations of **M**, as shown in Fig. R2, extending the work of Lee et al. Phys. Rev. B 108, 045132 (2023) <https://doi.org/10.1103/PhysRevB.108.045132>, where azimuthal dependence of the anisotropy has not been discussed. Our calculations show that projection onto the $\overline{\Gamma K}$ direction (as in our manuscript) is preferred. The in-plane anisotropy of ≈ 0.7 meV per formula unit is smaller than the out of plane anisotropy but significant, however, this number does not automatically translate into the spin reorientation transition temperature. This temperature is controlled by the collective behaviour of many local moments. In a very crude approximation, one needs to sum the Dy energies along some characteristic exchange length (“activation region”), thus obtaining much higher energy, and consequently much higher temperature. Furthermore, in real materials the existence of the surface and various shape anisotropies cannot be neglected, these have been considered neither in the results shown in Fig. R2 nor in the existing literature on R166 compounds.

Considering all the above, we find it plausible, that in the experiment we are probing 2 anti-aligned easy axis domains, with the **M** being at $\approx 45^\circ$ away from c-axis and projected along the $\overline{\Gamma K}$ azimuthal direction.

(2) There are two magnetization domains A and B, however the authors chose different integration regions for different analyses / binding energies, as marked in fig.1 (d) and (f). It is unclear why different integration regions are picked, or further - why isn't the whole area of a given domain integrated over for better signal. This needs to be explained, as it raises concerns of hidden spatial variability in the data.

This is a very understandable comment, and on the course of data analysis we have very carefully analyzed this.

Figures R3 and R4 show additional analysis we performed on our Dy 4*f* and Mn 3*p* data sets. One can see that the effects are uniform over the entire region (in panels (b) we show the curves for random pairs for regions indicated by squares in panels (a)) and that summing up the larger areas does not influence the conclusion.

Regarding Fig. 1**e**, these were measured at particular points (nominal beamspot size around $2\mu\text{m}$, cannot be much larger because of well-defined edges in Fig. 1**c**), because measurement of kx-ky maps at numerous locations would be exceedingly time consuming. Figure R5 shows the stability of the x-y micrograph over the entire measurement time, eliminating any possibility of drift.

Figures R3, R5, and R5 have been included in the revised Supplement as Figs. S6, S7, and S8, together with the discussion in Section SVII.

Figure R3: Confirmation of the generality of our results, the region of Dy 4f at $h\nu = 300$ eV. (a) x-y micrograph similar to Fig. 1d. (b) Curves such as the one in Fig. 2h but for the pairs of rectangles depicted in (a). (c) Same as (b), but sum of all rectangles in (a). The micrograph in (a) is plotted as difference of CD signals integrated over the regions indicated by vertical lines in (c).

Figure R4: Similar as Fig. R2, but for the Mn 3p spectral region at $h\nu = 300$ eV. Micrograph in (a) shows the CD difference between regions indicated in (c). Background correction (extracted from the low binding energy side of the spectra) has been applied in all cases.

Dy 4f 7F
 hv = 300 eV
 8PM July 1, 2024

Dy 4f 7F
 hv = 140 eV
 7AM July 2, 2024

Figure R5: Beam stability over the course of the measurements.

(3) The crystal orientation is rotated by 11 degrees compared to the analyzer's slit, as depicted in the experimental geometry shown in Fig. 1(b). Such orientation leads to substantial matrix element effects that conventional CD-ARPES experiments aim to mitigate by aligning the sample's high-symmetry direction with the slit. The two CD-Fermi surface maps presented in Fig. 4(c-d) exhibit a distinct two-fold structure around $k_y=0$, rather than the three- or six-fold texture that could be expected from the crystal structure's symmetry, indicating a major contribution of matrix element effect (in addition to final-state scattering/Daimon effect that the authors mention). The reason for choosing this sample orientation is unclear, especially since the authors go to great lengths to isolate the magnetic contribution from all the others. The effects of working with the plane-of-incidence of the light not along the mirror symmetry axis of the sample should be discussed.

This is another very clear and understandable comment.

There are different aspects that we need to discuss in relation to the above comment. At MAESTRO, one can either use the electrostatic lens deflector, or rotate the entire analyzer on differential seals. For this work we exclusively used the electrostatic deflector, which allows $\approx \pm 15^\circ$ "field of angular view". This was done to avoid any possible beamspot drift on the sample, which, although unlikely, may happen due to large weight shifts when rotating the analyzer physically. During our measurement, the cryogenic cooling has been stable since some days, and beamspot drifts have been negligible, as demonstrated in Fig. R5, which spans essentially entire 12h of measurements that led to the results presented in the manuscript. The measurement time at oversubscribed micro-ARPES beamlines is very limited, which was the main reason why the sample has not been rotated to align the reaction plane with one of the high-symmetry reciprocal directions. On the other hand, the generic orientation has advantages, because it allows to demonstrate the predictive power of our simulation approach, with spectacular near-quantitative agreements in Fig. 2b,c vs. Fig. 2f,g.

Figure 4c,d represent classical examples of CD-ARPES. We recently elaborated on CD-APRES in Phys. Rev. B 111, 115127 (2025) <https://doi.org/10.1103/PhysRevB.111.115127>. The dominant red-blue contrast stems predominantly from the Daimon effect. The slight rotation of the pattern stems from combination of Daimon effect with the geometrical mirror plane of the surface; this is because the Daimon effect reflects the crystallographic orientation of the sample. Additional contribution could be related to the inelastic mean free path (see Fig. 4 and

5 in Moser, JSERP Volume 214, 29 (2017), <https://doi.org/10.1016/j.elspec.2016.11.007>), but whether this effect differs from the Daimon effect is currently under the debate within the photoemission community.

The most important is physics. And from the point of view of physics and the message of the paper, we believe the azimuthal misalignment of the sample is not important.

Final minor suggestion: The red-to-blue color scale is used extensively throughout the manuscript and supplementary material, for related but distinct quantities. Choosing different colorscales for different quantities could improve the paper's readability.

We have changed the color convention for all panels that refer to spin polarization (we use violet-green for spin in the revised version).

We also added the descriptive text to panels Fig. 4c-d.

Our study deals with four quantities, two domains (*A* and *B*) and two light polarizations (C_+ and C_-). While we agree that the different linear combinations may appear confusing at first, we believe this is not an overwhelming complexity for a general audience reader.

Reviewer #2 (Remarks to the Author):

Reviewer #3 (Remarks to the Author):

This paper reports the magnetic properties of a Kagome lattice metal, DyMn6Sn6. By performing angle-resolved photoemission measurements of the Dy 4f and Mn 3p levels using a circularly polarized micro-focused beams with a diameter of less than 2 micrometer, the authors succeeded to reveal that the local magnetic moments of Dy and Mn exhibit ferrimagnetic ordering. The results obtained are interesting, but not surprising. Since even more interesting results could have been obtained with information on electron spin, I am wondering why the authors did not try to do spin-resolved photoemission measurements. Adding comments on this point would make the advantages of the method used by the authors in this study clearer.

We would like to thank Reviewer #3 for the positive report.

Visualization of the magnetic domains at this lateral scale requires microfocused ARPES. Currently, no microfocused ARPES facility combines the beamspot size of $\approx 2\mu\text{m}$ with spin-polarized measurements. For example, the Bloch beamline at MAXIV allows $10 \times 15 \mu\text{m}^2$ beamspot, and the NanoESCA beamline at Elettra has even a slightly larger beamspot.

Furthermore, SARPES measurements are very time consuming, therefore performing extensive exploratory mapping (as performed here) would not be feasible.

Furthermore, detailed explanation on why does the signs of $(I_{+A} + I_{-B})$ and $(I_{+B} + I_{-A})$ are opposite in the experimental result and theoretical one should be added, together with the reason of the difference in the percentage of CD when using different photon energies.

The quantities $I_{+A} + I_{-B}$ and $I_{+B} + I_{-A}$ have the same sign in experiment and in the theory for the Dy $4f$ and different signs in experiment and in the theory for Mn $3p$. This provides an experimental confirmation of the ferrimagnetic alignment of Dy and Mn local moments in DyMn_6Sn_6 . The quantities $I_{+A} + I_{-B}$ and $I_{+B} + I_{-A}$ are used in order to minimize the influence of the Daimon effect; we commented on that on page 3 of the main text and different aspects of the photoemission response due to non-zero OAM are discussed in Supplemental section SIV.

Minor points

Typo: On the first page, in the right column, “singatures” should be “signatures”.

Thank you for pointing out the typo, it has been corrected in the resubmitted version.

Reviewer #4 (Remarks to the Author):

The present manuscript has reported the study of the magnetic Kagome metal DyMn_6Sn_6 using high-resolution micro-focused circular-dichroic angle-resolved photoemission to probe its magnetic and electronic properties using combined experimental and theoretical techniques. The work can be very useful from experimental techniques point of view. I would definitely suggest this work to NCOMMS article. I still put one questions below whose entire descriptions not available.

We would like to thank the Reviewer #4 for positive comments.

Could you please elaborate the connecting techniques in supporting documents or in the main manuscript stating how ab-initio calculations were used for calculating photoemission matrix elements and which tool did you use?

There are two places in the manuscript where we discuss matrix elements.

- The details of calculations in Figs. **2f-g** and **3e-f** are described in the Section III C of the main text (Methods – Photoemission Calculations). First, the Dy $4f$ and Mn $3p3d$ multiplets have been calculated. Then one-electron photoemission matrix elements were calculated using the EDAC electron diffraction code (Ref. [15] in the main text), a broadly used tool that uses multiple scattering formalism. For this purpose we used a model cluster shown in Fig. R6. Finally, from the obtained one-electron matrix elements we constructed the many-electron matrix elements (Eq. 1 of the main text) for the multiplets of Dy and Mn.
- Further, the general description of the matrix elements from the $m \neq 0$ orbitals is included in the Supplemental section SIV. This description is pointing out that the Daimon scattering is suppressed, but not completely absent, if one plots quantities such as $I_{+A} + I_{+B}$ and $I_{-A} + I_{-B}$. This suppression allows to visualize the OAM-related signal in Fig. **4e-g** of the main text, which is otherwise obscured by the Daimon effect in Fig. **4c-d**. Our description of the matrix elements in the Supplemental section SIV is

following general rules of matrix elements and operators, without the use of any specific tool. For this reason, it is made for the atomic case, however, it allows to understand which contributions can be cancelled even if multiple scattering would be included. This justifies the comparison between the experimental map in Fig. 4g and the *ab-initio* (DFT LAPW) initial state calculation in 4k. Performing numerical one-step model photoemission calculation for the dispersive bands on the surface of the compound such as DyMn₆Sn₆ is a significant challenge, and such calculations are rarely (if ever) performed for such complex materials; it could be a topic of a future collaboration with one of the (few) one-step theory groups.

Figure R6: Model cluster used in the calculations of Dy 4f and Mn 3p photoemission. Red colour stands for Mn, blue – for Dy, grey – for Sn.